# Morphology and Vessel Density of the Macula in Preterm Children Using Optical Coherence Tomography Angiography

**DOI:** 10.3390/jcm11051337

**Published:** 2022-02-28

**Authors:** Agnieszka Czeszyk, Wojciech Hautz, Maciej Jaworski, Dorota Bulsiewicz, Justyna Czech-Kowalska

**Affiliations:** 1Department of Ophthalmology, Children’s Memorial Health Institute, 04-730 Warsaw, Poland; w.hautz@ipczd.pl; 2Department of Biochemistry, Radioimmunology and Experimental Medicine, The Children’s Memorial Health Institute, 04-730 Warsaw, Poland; m.jaworski@ipczd.pl; 3Department of Neonatology and Neonatal Intensive Care, Children’s Memorial Health Institute, 04-730 Warsaw, Poland; d.bulsiewicz@ipczd.pl (D.B.); j.kowalska@ipczd.pl (J.C.-K.)

**Keywords:** retinopathy of prematurity, optical coherence tomography angiography, retinal vessel density, foveal avascular zone, foveal hypoplasia, ganglion cell complex, choroidal thickness

## Abstract

Background: Retinal morphology changes may be associated with prematurity and can lead to visual impairment. Optical coherence tomography angiography may contribute to understanding the pathomechanism of structural and vascular retinal impairment in premature children. The aim of this study was to assess an influence of prematurity, neonatal clinical characteristics, and a history of retinopathy of prematurity (ROP) on the morphology and retinal vascularity of macula in children. Methods: A case–control study of 123 preterm children and 86 full-term children was performed. The age of the subjects was 10.45 years (IQR: 8.12–12.77), while the age of the control group was 11.78 years (IQR: 8.81–13.79). Optical coherence tomography (OCT) and optical coherence tomography angiography (OCTA, angio-OCT) were performed using RTVueXR Avanti (Optovue, Fremont, CA, USA). Ganglion cell complex (GCC), foveal thickness (FT), parafoveal thickness (PFT), size of foveal avascular zone (FAZ) in superficial plexus, superficial capillary vessel density (sVD), deep capillary vessel density (dVD), central choroidal thickness (CCT), and presence of macular hypoplasia were analyzed. The association between OCT/angio-OCT results and clinical characteristics including the degree of ROP and therapy requirements was assessed in preterm infants. Results: Foveal morphology was affected in preterm children with high incidence of foveal hypoplasia (24.77%). GCC was thinner in preterm children compared to controls: avgGG 93 μm vs. 100 μm, *p* < 0.001. No associations between GCC and gestational age (*R* = −0.085; *p* = 0.228) and birth weight (*R* = −0.054; *p* = 0.446) were found. FAZ in preterm group was smaller than in controls (0.13 ± 0.09 vs. 0.22 ± 0.09; *p* < 0.001). FAZ area correlated with gestational age (*R* = 0.456; *p* < 0.001) and birth weight (*R* = 0.472; *p* < 0.001). Deep vessel density in the fovea was higher in preterm children than in control group (*p* < 0.001). PFT was significantly lower in preterm children compared to control group. However, increased thickness in the fovea was noted in preterm children (*p* < 0.001). FT was inversely correlated with gestational age (*R* = −0.562; *p* < 0.001) and birth weight (*R* = −0.508, *p* < 0.001). CCT was lower in preterm children (312 μm vs. 337.5 μm, *p* < 0.001) Parameters of GCC and FT were higher in patients with ROP required treatment compared to patients without ROP and spontaneously regressed retinopathy. FAZ was smaller in patients with retinopathy than in preterm children without ROP. Conclusion: Prematurity has a significant negative impact on GCC, macular morphology, and vascularization. In premature children, decreased FAZ, increased FT, and vessel density were strongly associated with gestational age, birth weight, Apgar score, ROP stage, and treatment requirement. Optical coherence tomography angiography is a useful tool for detecting retinal changes in premature children.

## 1. Introduction

Retinopathy of prematurity (ROP) is vasoproliferative disorder that commonly occurs in preterm children and still remains an important cause of blindness in childhood worldwide. Retinal neovascularization develops due to production of vascular endothelial growth factor (VEGF) in response to the primary avascular retinal area [1,2,3,4]. The aim of currently available treatment is to decrease the level of VEGF directly by intravitreous injection of an anti-VEGF agent or to destroy the peripheral avascular retina with laser therapy or cryotherapy [5]. Visual outcome depends on ROP stage and treatment method. Children with a history of prematurity are reported to have higher risk of ocular disorders including reduced vision acuity, high refractive error, anisometropia, amblyopia, strabismus, glaucoma, cataract, and retinal detachment [1,2,3,6,7].

Children born preterm can have lower visual acuity despite normal-appearing fundus [8,9]. The differentiation and maturation of the fovea and retinal layers begin at 24 to 27 weeks after conception and finish at 8 months of age [10,11,12]. Preterm birth can lead to the disruption of normal retinal development. Abnormal foveal contour, foveal hypoplasia with absence of foveal depression, retention of inner retinal layers at foveal center, macula edema, retinoschisis, preretinal neovascularization, and retinal detachment have been reported in preterm children [2,3,13].

Optical coherence tomography (OCT) is a noninvasive technique which enables visualizing the structure of the retina, and dye-less optical coherence tomography angiography (OCTA, angio-OCT) additionally shows retinal and choroidal microcirculation [14,15]. OCTA enables a quantitative assessment of macular perfusion via detecting erythrocyte movement and provides high-resolution images of superficial and deep capillary plexuses of the foveal avascular zone [14,16]. OCT and OCT angiography allow measurements of the ganglion cell complex (GCC), retinal nerve fiber layers (RNFL), retinal thickness (RT), choroidal thickness (CT), foveal avascular zone (FAZ), and vessel density (VD) [14,17].

The ganglion cell complex is an indicator of nerve tissue damage. Assessment of progression of GCC changes is an essential tool in monitoring glaucoma or optic neuropathy and detecting optic nerve damage [18,19]. Ganglion cells are very sensitive to acute hypoxia, which causes their apoptosis [20]. Nerve tissue damage due to episodes of hypoxia during delivery or the neonatal period can affect the ganglion cell complex in premature children [21]. Changes in ganglion cell complex thickness have been reported in the literature in preterm children [22,23].

On the other hand, there are limited studies presenting choroidal changes including decreased thickness of the choroid in preterm children [24,25]. This finding is very important in the context of the choroid’s role in providing oxygen and nutrients to the external retina, being the only source of nutrition for the foveal avascular zone [26]. Additionally, there is a disrupted development of retinal vessels in preterm children. Vessel growth depends on oxygen gradient and chemical factors which are disturbed in prematurity. The critical period for foveal vasculature development is between 24 and 28 weeks of gestation. This is the reason why preterm birth can affect the foveal vasculature [27,28].

Changes in the vascular density and decreased size of the FAZ in preterm children compared to full-term controls have been reported in the literature [16,29,30,31].

The aim of the study was to assess an influence of prematurity, neonatal clinical characteristics, and a history of ROP on the morphology and retinal vascularity of macula in children using optical coherence tomography angiography (OCTA).

## 2. Material and Methods

This prospective, case–control study was conducted at the Department of Ophthalmology of The Children’s Memorial Health Institute in Warsaw between January 2016 and May 2021. The study was approved by the Bioethics Committee of The Children’s Memorial Health Institute (21/KBE/2019) and adhered to the tenets of the Declaration of Helsinki. Written informed consent was obtained from parents or the patient’s legal guardians and from patients ≥ 16 years old. Inclusion criteria for the study group were as follows: age ≤ 18 years and gestational age < 37 weeks. Exclusion criteria were as follows: ROP 4 and 5, as well as other concomitant retinal pathologies, such as hereditary retinal dystrophies, vitreoretinal diseases, media opacities including cataract preventing detailed imaging, history of uveitis, glaucoma, optic neuropathy, eye surgery, ocular trauma, or retinal detachment. Poor-quality scans due to low visual acuity or poor cooperation were also excluded. Analyzed scans had a signal strength index greater than 40 and no evidence of motion artefacts.

Control subjects were defined as having a normal ophthalmic examination and a best-corrected visual acuity (BCVA) of 0.0 according to logMAR chart, with no refractive error and no history of prematurity.

Data of neonatal characteristics were extracted from infants’ medical charts. Gestational age, birth weight, intrauterine growth restriction (IUGR) defined as birth weight < 10th percentile for gestational age, Apgar score, and stage of ROP including requirement of treatment were analyzed [32,33]. The indications for ROP treatment were based on the ETROP study [5].

All participants underwent a complete ophthalmological examination, including best-corrected visual acuity (BVCA) assessed with logMAR chart, anterior segment slit-lamp biomicroscopy, dilated fundus examination, and cycloplegic refraction using 1% tropicamide.

The spectral domain OCT (SD-OCT) and OCT angiography were performed using RTVue XR Avanti OCT with the AngioVue imaging system (Optovue, Fremont, CA, USA) and SSADA algorithm.

The GCC scan protocol consisted of one horizontal line with 7 mm scan length and 15 vertical lines with 7 mm scan length at 0.5 mm intervals centered at 1 mm temporally to the fovea. GCC thickness was measured as a distance between the internal limiting membrane (ILM) and the outer boundary of the inner plexiform layer (IPL) and was automatically calculated by the device (Figure 1). The RTVue XR Avanti measures GCC thickness separately in the superior (supGCC) and inferior (inf GCC) sector, as well as the average thickness in both sectors (avg GCC) [34].

A crossline scan was performed to obtain high-quality images of the retina and choroid. Choroidal thickness was measured manually, using a caliper in SD-OCT software. Choroidal thickness was defined as the distance between the hyperreflective line corresponding to the outer boundary of the retinal pigment epithelium (RPE) and the hyperreflective line corresponding to the chorioscleral border. The measurements were obtained in the subfoveal region, which was defined as the lowest point shown on the retinal images.

The presence of macular hypoplasia was analyzed on crossline scans according to classification presented by Thomas et al. [35].

OCTA was performed using 3 mm × 3 mm images of the macula centered on the foveola (Figure 2). Each OCTA en face image contains 304 × 304 pixels created from the intersection of 304 vertical and 304 horizontal B-scans [29]. AngioVue automatically segments the area into four layers: superficial capillary plexus layer (SP), deep capillary plexus layer (DP), outer retina layer, and choriocapillaries. Integrated automated algorithms provided by the machine software were used to quantify FAZ area (mm^2^) and macular vascular density (%). FAZ area was automatically calculated for superficial plexus [2,14]. The FAZ area is noted as an avascular region within the central fovea (Figure 3). It is measured at the thinnest point of the inner retina visible on OCTA B-scan with no vessels and blood flow, adjacent to the vascular area [3,10]. The foveal region is a 1 mm circle, while the parafoveal region is a ring with an internal diameter of 1 mm and external diameter of 3 mm [6]. Capillary vascular density in the macular and paramacular region was measured in both superficial and deep plexuses. Vessel density was calculated as the percentage area occupied by flowing blood vessels in the selecting region. The whole superficial capillary vessel density (wsVD), foveal superficial vessel density (fsVD), parafoveal superficial vessel density (psVD), whole deep vessel density (wdVD), foveal deep vessel density (fdVD), and parafoveal deep vessel density (pdVD) were analyzed.

Foveal thickness (FT) and parafoveal thickness (PFT) data were obtained from retinal maps, using the same device, and measured in micrometers (μm). Head stabilization was achieved with the standard chinrest and forehead support. Patients were asked to focus on an internal fixation target. Poor-quality scans, with motion artefacts or blurred images, were excluded [36]. The data collected from both eyes of the studied patients were analyzed.

## 3. Statistical Analysis

Nonparametric statistics were used. Data for categorical variables were presented as numbers and percentages, while continuous variables were presented as medians and interquartile ranges (IQRs). The Mann–Whitney test and Kruskal–Wallis test were used for comparisons of two and more than two groups, respectively. Correlation analysis was carried out by calculating Spearman’s *R*-value.

The subgroup analyses were performed in preterm children according to birth weight (<1500 g vs. ≥1500 g), gestational age (<28 weeks vs. ≥28 weeks), Apgar score (<4 in first minute), IUGR, ROP stage, and laser treatment requirement.

Statistica software v. 10 (StatSoft Inc., Tulsa, OK, USA) was used. A *p*-value less than 0.05 was considered significant.

## 4. Results

The clinical characteristics of the study group of 123 preterm children at the age of 10.45 years (IQR: 8.12–12.77) are presented in Table 1. A total of 203 eyes of 123 preterm children were included in the study, but 43 (17.5%) eyes were excluded from the analysis due to poor-quality images. Choroidal thickness was measurable in 179 eyes, while scans of 67 (27.2%) eyes were excluded due to movement artefacts. Good-quality OCT angiography images were available for 106 (43%) eyes of premature children. A comparison of clinical and ophthalmic characteristics between analyzed eyes and eyes excluded from the analysis in the study population of preterm children is presented in the Appendix A. The control group consisted of 86 healthy children (48 females, 38 males) at the age of 11.78 years (IQR: 8.81–13.79). A total of 171 eyes were analyzed. One eye was excluded due to poor-quality scans. Good-quality OCT angiography images were available for 133 (77.3%) eyes.

### 4.1. Macular Hypoplasia

Macular hypoplasia was present in 56 (24.77%) out of 226 eyes of preterm children. 20 eyes were excluded due to poor-quality scans. First-degree hypoplasia was noticed in 15 eyes (6.63%), second-degree hypoplasia was noticed in 11 eyes, (4.86%), third-degree hypoplasia was noticed in 20 eyes (8.84%), and fourth-degree hypoplasia was noticed in 10 eyes (4.42%). An irregular foveal contour with retinoschisis was noted in two eyes (0.88%).

### 4.2. Ganglion Cell Complex

The ganglion cell complex (regardless of location) was significantly thinner in preterm children than in healthy control group (Table 2). Additionally, GCC was thinner in patients with Apgar score < 4 and children with IUGR (Table 3). No correlations between GCC and gestational age (*R* = −0.085; *p* = 0.228) and birth weight (*R* = −0.054; *p* = 0.446) were found (Table 4). 

### 4.3. Vessel Density

Superficial whole vessel density, as well as that in the superior and inferior quadrants, was lower in preterm children, whereas deep vessel density in the fovea was significantly higher in preterm children than in healthy controls (Table 2).

There was a correlation between parafoveal superficial vessel density and birth weight (*R* = 0.298; *p* = 0.002) but not gestational age (*R* = 0.184; *p* = 0.059). Higher density in the fovea was associated with lower gestational age (superficial: *R* = −0.376; *p* < 0.001, deep: *R* = −0.444; *p* < 0.001) and lower birth weight (superficial: *R* = −0.306; *p* < 0.001, deep: *R* = −0.339; *p* < 0.001) (Table 4). Lower parafoveal and higher foveal superficial vessel density were noted in children < 1500 g (Table 3).

Whole superficial vessel density was lower in patients with Apgar score < 4 (*p* = 0.046).

Superficial vessel density of the fovea was higher in preterm children than in the control group, but the difference was not significant (Table 2). No associations were found in whole deep and parafoveal deep vessel density. No significant differences between gestational age and whole superficial (*R* = 0.051; *p* = 0.604), whole deep (*R* = −0.451; *p* = 0.645), and parafoveal vessel density (superficial: *R* = 0.184; *p* = 0.059, deep: *R* = 0.044; *p* = 0.065) were found (Table 4). Birth weight did not have an influence on whole and parafoveal deep vessel density (Table 3). Additionally, no significant difference was found in density in patients born < 28 weeks comparing to ≥28 weeks and with IUGR (Table 3).

### 4.4. Foveal Avascular Zone

FAZ in the preterm group was significantly smaller compared to control patients (*p* < 0.001). There was a positive correlation between FAZ and gestational age (*R* = 0.456; *p* < 0.001) and birth weight (*R* = 0.47; *p* < 0.001) (Table 4). FAZ was significantly smaller in children born < 28 weeks than in children born ≥ 28 weeks and in children with birth weight < 1500 g than ≥1500 g (Table 3). 

### 4.5. Retinal and Choroidal Thickness

Parameters of whole retinal thickness and parafoveal thickness were lower in preterm children than in healthy controls (Table 2). There was an inverse correlation between the foveal thickness and gestational age (*R* = −0.562; *p* < 0.001) and birth weight (*R* = −0.508; *p* < 0.001) (Table 4).

The retina was significantly thicker in the fovea of preterm children than healthy controls (Table 2). More immature children (gestational age: <28 weeks vs. ≥28 weeks) and with lower birth weight (<1500 g vs. ≥1500 g) had a thicker whole, foveal, and parafoveal retina (Table 3), but this was only statistically significant in the fovea (*p* < 0.001).

The choroid was significantly thinner in preterm children than in the healthy control group (312 vs. 337.5, *p* = 0.005) (Table 2).

### 4.6. The Influence of Stage of Retinopathy and Laser Treatment on OCT and OCTA Parameters

All examined eyes of preterm infants were divided into three groups regarding ROP history as follows: eyes without ROP (Group 0), eyes with ROP without treatment (Group 1), and eyes with ROP which required treatment (Group 2). Laser therapy was performed in all treated eyes; only one patient (two eyes) had laser therapy combined with ranibizumab injection. GCC was analyzed in 183 eyes. The remaining eyes had poor-quality images and, therefore, were excluded from the analysis. Good-quality OCTA scans were available for 89 eyes. The Kruskal–Wallis test showed a significantly thicker GCC in treated ROP eyes compared to eyes without ROP and with spontaneous regression of ROP (Table 5).

In OCT angiography, there was significant difference between eyes without ROP and treated ROP eyes in terms of superficial whole density, as well as foveal and parafoveal vessel density. Additionally, smaller deep foveal density was observed in eyes without ROP compared to eyes with regressed ROP. No associations in other parameters of vessel density between groups were found. Eyes following ROP treatment had higher foveal thickness compared to other eyes with ROP. Smaller FAZ was found in eyes with treated ROP in comparison to eyes without ROP or spontaneously regressed ROP (Table 5).

## 5. Discussion

It has been well documented that prematurity has a great impact on the development of retinal morphology and vascular network [1,2,3,6,7]. OCTA is a very useful, noninvasive method for visualization of the optic disc, retina, choroid, and their vasculature. It is very helpful in pediatric patients and enables detecting abnormalities that affect vision and cause amblyopia.

A critical period for macula development including the perifoveal vascular plexus, FAZ, and foveal pit formation is between 24 and 27 weeks post conception. The foveal pit is created due to migration of inner retinal neurons away from the fovea. FAZ forms in the process of apoptosis of foveal vessels. Preterm delivery disrupts this process and may have huge implications on visual acuity and retinal morphology [11,12,13].

The present prospective case–control study investigated the influence of prematurity, neonatal clinical characteristics, and diagnosis of ROP on macular morphology and retinal vascularity in preterm children. We found that the ganglion cell complex was thinner in preterm children compared to the control group; however, no associations with birth weight and gestational age were found. In contrast to our results, Yanni et al. and Rosen et al. found thicker GCC in preterm children [37,38]. Similarly to our study, Pueyo et al. noted ganglion cell damage in preterm children. GCC results were associated with low birth weight but not with gestational age [22]. Miki et al. also presented decreased GCC in preterm children compared to controls [23]. Neurological deficits but also the high retinal immaturity due to lower gestational age might result in differences in GCC data, and this requires further attention. Interestingly, the ganglion cell complex was thinner in patients with Apgar score < 4 and IUGR, which may suggest a negative impact of hypoxia and prolonged nutritional deficits during the vulnerable prenatal period of life. It might be speculated that severe hypoxemia at birth, as well as ROP, might disrupt further retinal vascularization with long-term consequences. As far as an association between ROP and Apgar score is concerned, it should be underlined that hypoxia at birth (Apgar score < 4) has been reported as a predictor of severe ROP [39,40,41]. On the other hand, other authors did not find any association between ROP including AP-ROP and Apgar score [42,43]. The significant limitations of Apgar score in the diagnosis of perinatal hypoxia in preterm infants should be underlined. It would be better to base the assessment on the more reliable markers of perinatal asphyxia as the result of an umbilical cord blood gas test. Unfortunately, such measurement was previously not a routine procedure in the delivery room. Therefore, the influence of low Apgar score on OCT/angio-OCT results should be interpreted with caution. The superficial vascular plexus begins to develop at around 21 weeks of gestation, while the deep vascular plexus begins to develop at around 25 to 26 weeks of gestation [44,45]. Therefore, prematurity has a great impact on the formation of both vascular plexuses. Lower superficial vessel density was noted in the preterm group compared to healthy controls but no difference in the deep vessel plexus was found. In addition to the fovea, increased vessel density was found in the superficial and deep plexus and was associated with lower gestational age and low birth weight. This might be the outcome of foveal immaturity and interruption of foveal apoptosis due to preterm birth. Similarly to our results, a significantly higher capillary density in both plexuses of the fovea was reported in the previous study [2,3,15].

Furthermore, FAZ area was smaller in preterm children than in healthy controls. Small FAZ was associated with low birth weight, lower gestational age, and a history of ROP. Apgar score did not have an impact on size of FAZ. Significantly decreased thickness of the choroid was noted in the preterm group. The choroid is the main source of nutrition for FAZ; hence, choroidal damage might implicate FAZ changes. In our study, FAZ was automatically calculated by software, whereas manual calculation is also used [15]. It should be underlined that no difference between manual and automated calculation of FAZ was reported [44,45,46]; thus, a comparison of such results is acceptable. Our study presented significantly smaller FAZ in preterm children than in the control group, similarly to the several previous reports [2,3,15,29,30]. The explanation for smaller FAZ in preterm babies is that the fovea is vascularized in the fetus. Retinal blood vessels are shown in the optic disc at 14 weeks of gestation and grow nasally and temporally, reaching the fovea at 22 weeks of gestation. They fully cover the macular area up to 25 weeks [28]. A further step of macular formation is vessel apoptosis to form the avascular zone after 36 weeks of gestation [13,41]. Therefore, children born before 30 weeks have smaller or absent FAZ. Children born after 36 weeks usually have normal FAZ [28,47]. A small FAZ might be a sign of prematurity in adults [47]. Balasubramanian et al. suggested that a small FAZ could be an indicator of worse visual outcomes of preterm babies [2]. Furthermore, in our study, FAZ changes were associated with lower visual acuity.

The absence of FAZ was typically associated with foveal depression and foveal hypoplasia. We included preterm children with foveal hypoplasia; however, in all cases, FAZ was measurable. In our study, almost 25% of preterm children had macular hypoplasia. Patients with third- and fourth-degree foveal hypoplasia did not manage to fixate properly to obtain good-quality scans; thus, measurement of FAZ was technically impossible.

Vinekar et al. did not notice differences in sVD and dVD according to birth weight, whereas FAZ was smaller in patients with birth weight < 1500 g [15]. Our results allow a similar conclusion regarding FAZ area, but we noted different parameters in foveal and perifoveal vessel density between groups.

Central choroidal thickness was lower in premature children similarly to previous reports [24,26]. Furthermore, retinal thickness in the fovea was higher in preterm children, which was also confirmed by other authors [2,3,29].

Main treatment methods for ROP currently available are anti-vascular endothelial growth factor (VEGF) intravitreal injections and laser photocoagulation [5]. Risk of delayed retinal maturation leading to a large area of the avascular zone is a possible side-effect of anti-VEGF treatment [48,49]. On the other hand, laser therapy is associated with risk of high myopia and glaucoma in patients and leads to peripheral visual field deficits in the area of laser spots [4,50]. The disadvantage of OCT and OCTA is the visualization of only the posterior pole and a lack of assessment of the peripheral retina following ROP changes. The negative influence of laser therapy on macular area in eyes following ROP treatment has been recently described [2,3,8,15]. In subgroup analysis based on ROP severity, we found a significantly thicker fovea in patients who required treatment. Additionally, we found increased GCC and foveal thickness in patients with ROP requiring laser therapy compared to the group without ROP or spontaneous ROP regression. This might suggest the severe immaturity of retinal layers in patients with more advanced ROP stages. FAZ area was smaller in all our ROP patients compared to children without ROP. Our findings are consistent with previous reports presenting a significant association between laser therapy and smaller FAZ and increased foveal thickness [2,3,8,15].

The present study compared OCT and angio-OCT parameters between preterm children and control full-term born children. The strength of our study is the large cohort consisting of 209 participants including 123 preterm children. The additional value is a comparison between subgroups based on neonatal characteristics. To the best of our knowledge, this is the first study widely describing comparisons of OCT/angio-OCT parameters in patients without ROP, spontaneously regressed ROP, and severe ROP in a population of preterm children. However, this study also had some limitations. Firstly, a quite large number of eyes were excluded from the analysis of angio-OCT parameters due to poor quality or lack of cooperation across preterm children. Children with a history of prematurity have difficulty obtaining proper fixation because of neurological deficits, low visual acuity, or nystagmus. Nevertheless, most GCC scans and crossline scans were successfully performed due to the short examination time, but the excluded eyes were those with worse pathology, poorer visual acuity, and higher refractive error. Additionally, lower gestational age and birth weight, as well as a higher prevalence of ROP and ROP treatment, were noted in excluded eyes than analyzed eyes in the case of GCC images. Additionally, younger patients with lower gestational age and ROP had a higher percentage of excluded OCTA images, which are more demanding and time-consuming to obtain. It might be expected that wider analysis with excluded eyes might strengthen our result or help to reveal differences in the other examined parameters of OCT/angio-OCT rather than cease proven differences between preterm children and healthy controls. However, OCTA parameters should be interpreted with caution because the selection bias cannot be neglected due to the higher percentage of excluded eyes from the analysis. Secondly, AngioVue software did not use skeletonized vessel density which might provide a more accurate assessment. Currently available handheld OCT angiography will hopefully provide additional value in the diagnosis and management of preterm children to gain more insight into the mechanism of ROP. Thirdly, the study group had a limited number of patients with aggressive posterior retinopathy of prematurity (AP-ROP) and plus disease; therefore, a separate analysis for these subgroups was not performed. Lastly, the study included only one patient following anti-VEGF injection, because this treatment was introduced later. Future OCTA studies are needed in a larger group of ROP patients following anti-VEGF therapy to compare both methods of treatment.

## 6. Conclusions

OCTA seems to be a very promising tool to diagnose and monitor retinal changes in preterm children because of the quick time exposure and noninvasiveness; hence, it should be performed routinely. However, the limitation of the OCTA technique is the difficulty in obtaining good-quality images for analysis caused by fixation problems more often present in this specific population.

This study confirmed the influence of prematurity on FAZ size, vessel density, and retinal and choroidal thickness. We documented the association of selected neonatal parameters with macular morphology and vascularity. This study showed that the impairment of retinal structure and vessel density is more prominent in patients with severe ROP requiring laser therapy. Further studies are needed to explore the microvasculature in preterm children and choroidal changes to gain more insight into the pathomechanism of retinopathy with respect to structural and functional outcome.

## Figures and Tables

**Figure 1 jcm-11-01337-f001:**
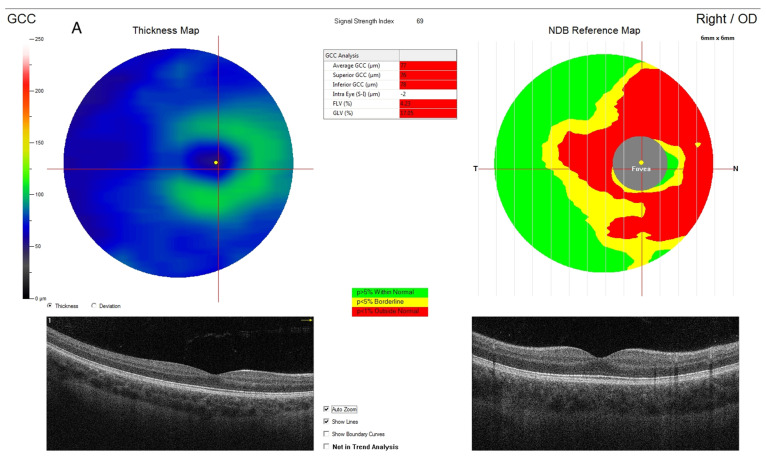
Measurement of retinal ganglion cell complex thickness in preterm 12 year old boy (30 weeks gestational age, birth weight 1200 g, no history of ROP) with decreased parameters (**A**) and in healthy full-term born child with normal parameters (**B**).

**Figure 2 jcm-11-01337-f002:**
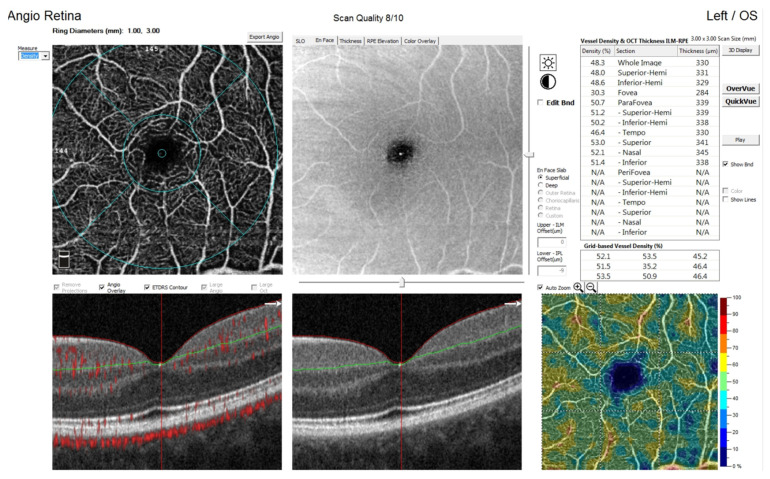
Image presenting superficial vessel density measurements.

**Figure 3 jcm-11-01337-f003:**
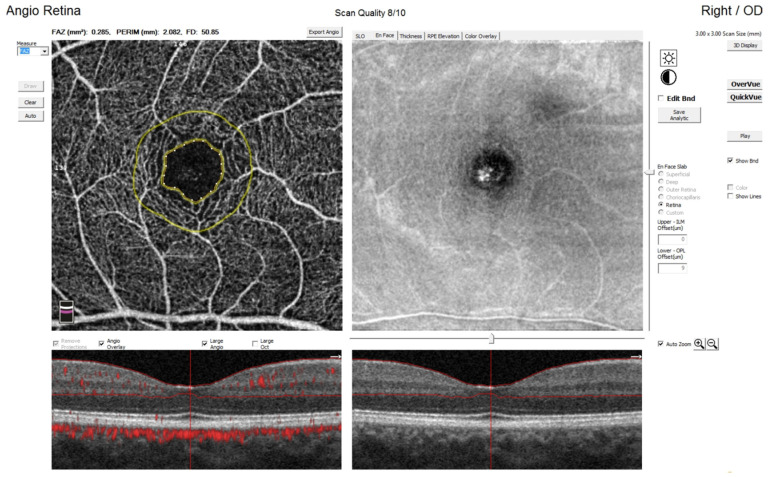
Foveal avascular zone (FAZ) measurement in superficial capillary plexus.

**Table 1 jcm-11-01337-t001:** Demographic and clinical characteristics of preterm children. Data are presented as a numbers and percentages (%) or medians and interquartile ranges (IQR).

Parameters	Number (%) or Median (IQR)
Preterm children	123
Males	66 (53.66%)
Age (years)	10.58 (8.12–12.77)
Birth weight (g)	1350 (930–1678)
Birth weight < 1500 g	75 (62.5%)
IUGR	14 (11.38%)
Gestational Age (weeks)	29 (27–32)
Gestational Age < 28 weeks	35 (28.46%)
Apgar score	
0–3	23 (18.7%)
4–7	67 (54.47%)
8–10	22 (17.89%)
Lack of data	11 (8.94%)
Ophthalmic characteristics	
**Parameter**	**Number of eyes (%)**
No ROP	85 (34.55%)
ROP	129 (52.44%)
ROP 1 stage	9 (6.98%)
ROP 2 stage	57 (44.19%)
ROP 3 stage	63 (48.83%)
Lack of ROP data	32 (13.01%)
Plus disease	47 (36.43%)
ROP with treatment	70 (54.26%)
Visual Acuity (logMAR)	0 (0.22–0)
Refractive error (Dioptres)	1.0 (−0.75–2.27)
Macular hypoplasia	56 (24.77%)
Lack of data	20

**Table 2 jcm-11-01337-t002:** Comparison of ganglion cell complex thickness, choroidal thickness, and vessel density parameters between preterm children and healthy control subjects. Data are presented as medians and interquartile ranges (Q1–Q3).

Variable	Preterm Children	*N*	Healthy Control	*N*	*p*-Value
Average GCC (μm)	93 (86–100)	203	100 (96–103)	165	<0.001
GCC superior quandrant	92 (85–98)	203	99 (94–104)	165	<0.001
GCC inferior quadrant	93 (87–101)	203	100 (96–104)	165	<0.001
CCT (μm)	312 (250–360)	179	337.5 (280–381)	130	0.005
FAZ (μm^2^)	0.13 (0.07–0.19)	106	0.23 (0.17–0.3)	133	<0.001
wsVD	45.2 (41.7–48)	106	48.7(47–50.3)	133	<0.001
wsVD superior quadrant	45.1 (41.8–47.6)	106	48.7(47–50.2)	133	<0.001
wsVD inferior quadrant	45.5 (42.1–48.2)	106	49.2 (46.9–50.5)	133	<0.001
fsVD	24.95(19.4–31.7)	106	23.9 (19.7–28.7)	133	0.398
psVD	47.75 (43.8–50.1)	106	51.4 (49.7–53)	133	<0.001
psVD superior quadrant	47.7 (43.7–50.4)	106	51.4 (49.6–53.3)	133	<0.001
psVD inferior quadrant	48.05 (49.3–50.3)	106	51.7 (49.5–53.3)	133	<0.001
wdVD	49.8 (47–52.3)	106	49.8 (46.5–52.7)	133	0.707
wdVD superior quadrant	50.1 (47.7–52.2)	106	49.8 (46.5–52.7)	133	0.644
wdVD inferior quadrant	49.35 (46.6–52.2)	106	49.5 (46.4 −52)	133	0.793
fdVD	41.9 (36.6–46.4)	106	36.9 (51.5–41.5)	133	<0.001
pfdVD	50.8 (48.4–53.7)	106	51.4 (48.9–54.0)	133	0.522
pdVD superior quandrant	51.6 (48.4–54)	106	51.15 (47.75–53.85)	133	0.706
pdVD Inferior quadrant	50.5 (47.9–53.8)	106	51.4 (48.5–54.1)	133	0.454
WT (μm)	310.5 (303–326)	106	321 (312–329,5)	133	<0.001
WT superior quadrant (μm)	313.5 (303–329)	106	323.5 (313–331)	133	<0.001
WT inferior quadrant (μm)	310 (303–325)	106	319.5 (311–329)	133	<0.001
FT (μm)	274.5 (257–298)	106	252 (242–268)	133	<0.001
PFT (μm)	318.5 (312–336)	106	331 (322–339)	133	<0.001
PFT superior quadrant (μm)	320.5 (311–336)	106	331 (322–339)	133	<0.001
PFT inferior quadrant (μm)	319 (312–335)	106	329.5 (321–338)	133	<0.001

*N*, number of eyes; GCC, ganglion cell complex; CCT, central choroidal thickness; FAZ, foveal avascular zone; wsVD, whole vessel density in superficial plexus; fsVD, foveal vessel density in superficial plexus; psVD, parafoveal vessel density in superficial plexus; WT, whole thickness; FT, foveal thickness; PFT, parafoveal thickness; wdVD, whole vessel density in deep capillary plexus; fdVD, foveal vessel density in deep capillary plexus; pdVD, parafoveal vessel density in deep capillary plexus.

**Table 3 jcm-11-01337-t003:** The results of OCT according to neonatal characteristic (gestational age, birth weight, Apgar score, IUGR) in preterm children.

	Gestational Age (Weeks)			Birth Weight (Grams)			Apgar Score (Points)			IUGR		
	Weeks	Median (IQR)	*p*-Value	Weight	Median (IQR)	*p*-Value	Apgar Score	Median (IQR)	*p*-Value	IUGR	Median (IQR)	
Avg GCC (μm)	<28 *N* = 47	96 (87–103)	0.059	<1500 *N* = 114	93 (87–100)	0.522	<4 *N* = 32	89.5 (82–93)	0.003	Yes *N* = 25	89 (80–93)	0.044
	≥28 *N* = 156	92 (86–98)		≥1500 *N* = 83	93 (86–98)		≥4 *N* = 150	95 (88–100)		No *N* = 174	94 (87–100)	
FAZ (μm^2^)	<28 *N* = 18	0.08 (0.04–0.15)	0.026	<1500 *N* = 55	0.08 (0.05–0.14)	<0.001	<4 *N* = 15	0.13 (0.05–0.19)	0.975	Yes *N* = 19	0.106 (0.074–0.208)	0.796
	≥28 *N* = 88	0.14 (0.07–0.2)		≥1500 *N* = 45	0.1 (0.14–0.25)		≥4 *N* = 80	0.1 (0.07–0.18)		No *N* = 83	0.141 (0.067–0.197)	
wsVD (%)	<28	44.65 (38.3–48.5)	0.604	<1500	44.6 (40.3–47.6)	0.116	<4	42.3 (38.8–46.1)	0.046	Yes *N* = 19	44 (40.4–45.3)	0.057
	≥28	45.25 (42.6–47.9)		≥1500	46.2 (43.2–48)		≥4	49.3 (46.5–51.95)		No *N* = 83	45.6 (42.3–48.1)	
fsVD (%)	<28	30 (23.5–32.8)	0.059	<1500	26 (21.5–32.8)	<0.001	<4	23.1 (16.4–32.2)	0.571	Yes *N* = 19	22.2 (16–27)	0.184
	≥28	24.15 (18.35–30.3)		≥1500	21.5 (16.3–27.4)		≥4	42.25 (36.8–47.05)		No *N* = 83	25 (19,4–31,6)	
psVD (%)	<28	45.55 (38.6–50.1)	0.222	<1500	46.5 (42–49.1)	0.011	<4	45.2 (40–48.7)	0,060	Yes *N* = 19	47 (43.4–48.7)	0.220
	≥28	47.95 (44.5–50.2)		>1500	49 (46.2–50.3)		≥4	47.45 (44.35–50.35)		No *N* = 83	48.5 (44.3–50.3)	
WT (μm)	<28	326.5 (318–347)	<0.001	<1500	317 (302–322)	0.231	<4	312 (295–344)	0.543	Yes *N* = 19	304 (296–318)	0.052
	≥28	308.5 (302–324)		≥1500	317 (302–332)		≥4	31 (306–328)		No *N* = 83	316 (304–329)	
FT (μm)	<28	314 (273–329)	<0.001	<1500	293 (266–315)	<0.001	<4	281 (262–308)	0.431	Yes *N* = 19	273 (256–287)	0.397
	≥28	271 (256–288)		≥1500	257 (252–278)		≥4	274.5 (257–298)		No *N* = 83	277 (256–300)	
PFT (μm)	<28	334.5 (323–355)	0.002	<1500	323 (310–339)	0.24	<4	318 (302–350)	0.446	Yes *N* = 19	314 (304–330)	0.117
	≥28	317.5 (310–332)		≥1500	323 (310–339)		≥4	322.5 (316–336.5)		No *N* = 83	323 (313–338)	
wdVD (%)	<28	48.7 (45.6–50.9)	0.292	<1500	49.3 (46.4–52.2)	0.526	<4	50.4 (47.5–54.3)	0.226	Yes *N* = 19	50 (48.8–54.2)	0.470
	≥28	50 (47.45–52.5)		≥1500	50.1 (47.5–52,5)		≥4	49.3 (46.5–51.95)		No *N* = 83	49.4 (46.7–52.3)	
fdVD (%)	<28	43.4 (40.3–49)	0.064	<1500	43.4 (40.3–48.9)	<0.001	<4	40,4 (35–48.4)	0.578	Yes *N* = 19	41.6 (32.9–44.7)	0.367
	≥28	41.65 (35.55–45.65)		≥1500	38.7 (32.9–43.8)		≥4	42.25 (36.8–47.05)		No *N* = 83	42.2 (36.6–47)	
pdVD (%)	<28	49.65 (47–52.6)	0.209	<1500	50.1 (47.2–53.6)	0.124	<4	52.6 (49.7–55.5)	0.101	Yes *N* = 19	51.6 (50–55.5)	0.359
	≥28	41.6 (48.5–54.15)		≥1500	51.7 (49.3–54.4)		≥4	50.3 (48.25–53.3)		No *N* = 83	50.4 (48.4–53.6)	

*N*, number of eyes; IUGR, Intrauterine Growth Restriction; avgGCC, average ganglion cell complex; FAZ, foveal avascular zone; wsVD, whole vessel density in superficial plexus; fsVD, foveal vessel density in superficial plexus; psVD, parafoveal vessel density in superficial plexus; WT, whole thickness; FT, foveal thickness; PFT, parafoveal thickness; wdVD, whole vessel density in deep capillary plexus; fdVD, foveal vessel density in deep capillary plexus; pdVD, parafoveal vessel density in deep capillary plexus; IQR, interquartile range (Q1–Q3).

**Table 4 jcm-11-01337-t004:** The Spearman correlations between OCT/OCTA parameters and gestational age and birth weight.

	Gestational Age			Birth Weight		
Parameter	*N*	*R* Spearman	*p*-Value	*N*	*R* Spearman	*p*-Value
AvgGCC	203	−0.085	0.228	199	−0.054	0.446
wsVD	106	0.051	0.604	102	0.195	0.049
fsVD	106	−0.376	<0.001	102	−0.306	<0.001
psVD	106	0.184	0.059	102	0.298	0.002
wdVD	106	−0.451	0.645	102	0.013	0.898
fdVD	106	−0.444	<0.001	102	−0.339	<0.001
pdVD	106	0.044	0.065	102	0.089	0.372
FAZ	106	0.456	<0.001	102	0.472	<0.001
WF	106	−0.389	<0.001	102	−0.186	0.06
FT	106	−0.562	<0.001	102	−0.508	<0.001
PFT	106	0.465	<0.001	102	−0.185	0.063

*N*, number of eyes; avgGCC, average ganglion cell complex; FAZ, foveal avascular zone; wsVD, whole vessel density in superficial plexus; fsVD, foveal vessel density in superficial plexus; psVD, parafoveal vessel density in superficial plexus; WT, whole thickness; FT, foveal thickness; PFT, parafoveal thickness; wdPD, whole vessel density in deep capillary plexus; fdVD, foveal vessel density in deep capillary plexus; pdVD, parafoveal vessel density in deep capillary plexus.

**Table 5 jcm-11-01337-t005:** Comparisons of OCT/OCTA results in examined eyes according to diagnosis of ROP and ROP severity in the study population of preterm children. Data are presented as medians (IQRs).

	Group 0 without ROP	Group 1 ROP without Treatment	Group 2 –Severe ROP Requiring Treatment	*p*-Value for Overall Kruskal–Wallis Test	*p*-Value Group 0 vs. Group 1	*p*-Value Group 0 vs. Group 2	*p*-Value Group 1 vs. Group 2
GCC	*N* = 85	*N* = 57	*N* = 41				
Avg GCC	93 (87–98)	91 (84–95)	100 (92–105)	<0.001	0.272	0.006	<0.001
OCT Angiography	*N* = 50	*N* = 21	*N* = 18				
FAZ (μm^2^)	0.16 (0.11–0.2)	0.09 (0.06–0.13)	0,09 (0.04–0.13)	0.003	0.046	0.009	1.0
wsVD	46 (43.2–48.1)	44.7 (50.5–48.2)	41.7 (38.3–45.4)	0.027	1.0	0.022	0.361
fsVD	22.1 (16.3–26.7)	26.2 (21–31.7)	31.55 21.5–33.7)	0.009	0.113	0.017	1.0
psVD	49.05 (46.4–50.9)	47.4 (41.5–50.3)	44.25 (38.6–47.5)	0.002	0.288	0.002	0.338
WT	310.5 (304–325)	307 (302–324)	325 (300–347)	0.187	1.0	0.339	0.261
FT	261.5 (255–279)	273 (257–298)	307 (290–321)	<0.001	0.442	<0.001	0.017
PFT (μm)	51.4 (48.6–54.8)	316 310–332)	332.5 (307–355)	0.250	1.0	0.688	0.296
wdPD	50.5 (47.4–52.3)	49.2 (46.6–52.2)	48.55 (45.9–52.2)	0.823	1.0	1.0	1.0
fdVD	40.2 (39.2–42.9)	44.3 (42–47.9)	42.3 (37–48.9)	0.004	0.009	0.073	1.0
pdVD	51.4 (48.6–54.8)	50.1 (47.8–53)	49.65 (47–55.4)	0.476	1.0	0.869	1.0

*N*, number of eyes; ROP, retinopathy of prematurity; avgGCC, average ganglion cell complex; FAZ, foveal avascular zone; wsVD, whole vessel density in superficial plexus; fsVD, foveal vessel density in superficial plexus; psVD, parafoveal vessel density in superficial plexus; WT, whole thickness; FT, foveal thickness; PFT, parafoveal thickness; wdVD, whole vessel density in deep capillary plexus; fdVD, foveal vessel density in deep capillary plexus; pdVD, parafoveal vessel density in deep capillary plexus; *p*-value < 0.05 considered statistically significant.

## Data Availability

Data are available on request due to restrictions.

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
