# Peer review of "Morphology and Vessel Density of the Macula in Preterm Children Using Optical Coherence Tomography Angiography"

_jcm, 2022, doi:10.3390/jcm11051337_

Round 1

Reviewer 1 Report

I found the article really interesting, the authors focused on the hot topic of the relevance of OCTA evaluation in ROP children
They assessed in a very accurate manner the correlation between the influence of prematurity on macular morphology and vascularity, with detailed attention on each OCTA parameter correlated with children clinical and demographic settings.
Several articles have been published on this topic in the last years and the Authors’ results seem to agree with them.
Here some suggestions that in my opinion, could give a more complete understanding of the subject:
- 1 ROP interferes with the normal development of the fovea that depends on the interaction between neurosensory retina and vascular growth; a brief explanation of this embriological process could help the reader to understand the normal and deseased FAZ anatomy and
diameter.
- 2 The Authors correctly admit the limitation of the study, in particular the difficult acquisition modalities of OCTA in children with nystagmus, poor visual acuity or optic media opacities, do they think that this could cause an involuntary exclusion of more pathologic eyes or more
abnormal data?
- 3 Did you find any distinctive OCTA features in ROP -PLUS disease?
- 4 Moshiri and colleagues published in 2019 on the feasability of a OCTA/ROP correlation study using a handheld device, do the Authors have any experience of this device and do they consider
this a future available option to examine ROP patients?
- 5 interestingly , reviewing the Literature , this is the first OCTA study in wich ROP features were directly correlated not only with Gestational age ad bird weight but also with Apgar score.
- 6 line 85, 144 and 147: please provide bibliograhy to support these sentences
- 7 line 105: please extend your discussion on correlation between GCC thickness and Apgar score, explain how this score is directly affected by ROP

Author Response

20 February 2022

Re:

Manuscript ID: jcm-1574815

Title: Morphology and vessel density of the macula in preterm children using Optical Coherence Tomography Angiography.

We would like to thank the reviewers for their constructive comments and suggestions, which enable us to substantially improve our manuscript.  Please find our response to comments below:

-1 ROP interferes with the normal development of the fovea that depends on the interaction between neurosensory retina and vascular growth; a brief explanation of this embriological process could help the reader to understand the normal and deseased FAZ anatomy and diameter.

Reply: We really appreciate the reviewer’s suggestion. We have done our best to follow above mentioned proposal and we have added the following sentence into the Discussion section: “Retinal blood vessels are shown at the optic disc at 14 weeks of gestation and grow nasally and temporally reaching fovea at 22 weeks of gestation. They cover fully macular area up to 25 weeks. Further step of macular formation is vessel apoptosis to form avascular zone after 36 weeks of gestation.  Therefore, children born before 30 weeks have smaller or absent FAZ, Children born after 36 weeks usually have normal FAZ”.

- 2 The Authors correctly admit the limitation of the study, in particular the difficult acquisition modalities of OCTA in children with nystagmus, poor visual acuity or optic media opacities, do they think that this could cause an involuntary exclusion of more pathologic eyes or more abnormal data?

The most scans of GCC (82%) and crossline (72.8%) were successfully performed in children with poor visual acuity and nystagmus and were included into the analysis. OCT angiography requires longer fixation and cooperation, so in these children, too many artifacts made it impossible for reliable analysis. Indeed, the percentage of excluded eyes was very high (57%) therefore the selection bias can’t be neglected. However, that does not mean that the examination is not useful in these children.  In individual cases, despite a slight nystagmus and low visual acuity, it was possible to perform good quality images that were used for analysis. It might be stressed that age of excluded eyes was comparable to analyzed eyes regarding GCC and CCT. Nevertheless, the excluded cases are those with more pathology and poorer VA, higher refractive error and less fixation in case of GCC images. Additionally gestational age, birth weight, presence of ROP and treatment played an important role in GCC analysis between included and excluded eyes. Younger patients with lower gestational age and retinopathy had higher percentage of excluded OCTA images. It might be expected that wider analysis with excluded eyes might strengthen our result or help to reveal differences also in other examined parameters of OCT/angio OCT rather than cease proven differences between preterm children and healthy controls.

We have performed comparison between the included and the excluded eye regarding baseline clinical and ophthalmological characteristics as recommended by reviewer #2. We have added these results into the manuscript as a supplementary file (Table S1). Interestingly, the statistical results are not fully consistent. As might be assumed same differences between excluded and included cases exist. We have addressed this issue in the Discussion section as an additional limitation of the study.

- 3 Did you find any distinctive OCTA features in ROP -PLUS disease?

Children with ROP – plus disease were included in group of children required treatment where with higher foveal thickness, vessel density and smaller size of FAZ comparing to other premature children.  The group of children with ROP-PLUS disease was too small to perform the separate statistical analysis. We have addressed this issue in the Discussion section as an additional limitation of the study: “Third, the study group have a limited number of patients  with aggressive posterior retinopathy of prematurity (AP-ROP) and plus disease therefore the separate analysis for these subgroups was not performed.”

- 4 Moshiri and colleagues published in 2019 on the feasibility of a OCTA/ROP correlation study using a handheld device, do the Authors have any experience of this device and do they consider this a future available option to examine ROP patients?

Reply: We do have experience with Leica Hand-held OCT, which we found useful in assessing macular oedema or epiretinal membrane but unfortunately, we do not have hand-held OCT angiography. We believe that the use of that device would be beneficial for children, especially preterm with retinopathy and help with management. Therefore, we are planning to buy this equipment this year and present the results in the future. We have addressed this issue in the Discussion: “Hopefully, currently available hand-held OCT angiography will give an additional value in diagnosis and management of preterm children to get more insight into mechanism of ROP”.

- 5 interestingly, reviewing the Literature, this is the first OCTA study in which ROP features were directly correlated not only with Gestational age ad bird weight but also with Apgar score.

Reply: Indeed, to our best knowledge this is the first study showing correlation between ROP retinal changes and Apgar score. However, we have found the negative influence of very low Apgar score (<4) only for average GCC and wsVD. It might be speculated that sever hypoxemia at birth might disrupt further retinal vascularization with long term consequences.

- 6 line 85, 144 and 147: please provide bibliography to support these sentences

Reply: We are grateful for this important comment. We have provided additional bibliography as you suggested.

- 7 line 105: please extend your discussion on correlation between GCC thickness and Apgar score, explain how this score is directly affected by ROP

Reply: We agree that this issue might be interesting for readers therefore the discussion has been extended as suggested. Unfortunately, there is not much information in the literature providing explanation for the influence of Apgar on GCC thickness. We are able only to provide suggestions based on our study and literature review regarding GCC. We have addressed this issue in the Discussion section: “it might be speculated that sever hypoxemia at birth as well as ROP might disrupt further retinal vascularization with long term consequences. As far as an association between ROP and Apgar score is concerned it should be underline that the hypoxia at birth (Apgar score<4) was reported as a predictor of severe ROP. On the other hand, other authors did not find any association between ROP including AP-ROP and Apgar score. The significant limitations of Apgar score in the diagnosis of perinatal hypoxia in preterm infants should be underlined.  It would be better to base the assessment on more reliable markers of perinatal asphyxia as the result of umbilical cord blood gases test. Unfortunately, previously such measurement was not a routine procedure at the delivery room. Therefore, the influence of low Apgar score on OCT/angio OCT results should be interpreted with caution”.

Reviewer 2 Report

Many thanks for your manuscript discussing morphology and vessel density in macula of preterm children using OCTA.

the content is genuine and novel and of interest to the readers and will definitely require further studies to validate your results 

My points for discussions are

  1. line 147: you need to consider using skeletonised vessel density as previously described in other papers reviewing vessel density analysis in OCTA. this would be more accurate and exclude the factor for vessel width in estimating vessel density. this parameter would require automated software to extract it rather than the machine segmentation and analysis only.
  2. line 159: Statistical analysis; the analysis is slightly weak. you used repeat M-W tests for multiple comparisons.....and inevitably you would get positive results with the repeated test and some comparisons would appear statistically significant. alternatively, did you consider regression analysis models which could be more accurate to verify the relationships. few models would be applied for unrelated variables. an expert statistical advice would be helpful to strengthen your analysis.
  3. line 172: I would like to see some analysis for the excluded eyes regarding their gestational age, BCVA, etc,......as it is likely that these cases are those with more pathology and poorer VA and less fixation and hence led to selection bias. this needs to be clarified comparing the patients characteristics between included and excluded eyes to exclude bias. 

Author Response

20 February 2022

Re:

Manuscript ID: jcm-1574815

Title: Morphology and vessel density of the macula in preterm children using Optical Coherence Tomography Angiography.

We would like to thank the reviewers for their constructive comments and suggestions, which enable us to substantially improve our manuscript.  Please find our response to the comments below:

the content is genuine and novel and of interest to the readers and will definitely require further studies to validate your results 

Reply: We appreciate positive attitude to the content of our manuscript and agree that further studies are required. We also plan to continue our study to increase the quantity of the study sample to extend statistical analysis. We would like to recruit more patients with AP-ROP and patients after anti-VEGF treatment.

1. line 147: you need to consider using skeletonised vessel density as previously described in other papers reviewing vessel density analysis in OCTA. this would be more accurate and exclude the factor for vessel width in estimating vessel density. this parameter would require automated software to extract it rather than the machine segmentation and analysis only.

Reply: Unfortunately, AngioVue software available in our Department is not having skeletonized option so we could not use it for our analysis. To be exhaustive we decided to add this information into the limitations of the study: “AngioVue software did not use skeletonized vessel density which might provide more accurate assessment”.

2. line 159: Statistical analysis; the analysis is slightly weak. you used repeat M-W tests for multiple comparisons.....and inevitably you would get positive results with the repeated test and some comparisons would appear statistically significant. alternatively, did you consider regression analysis models which could be more accurate to verify the relationships. few models would be applied for unrelated variables. an expert statistical advice would be helpful to strengthen your analysis.

Reply: We have discussed our results in the context of abovementioned suggestions with experienced statistician. However, this investigation has an exploratory nature, as some of the parameters have not been investigated previously in preterm children, so we focused on finding parameters that differentiate eyes with retinopathy from others. Since correction for multiple testing could have obscured these differences, authors decided to not carried out such procedure. The suggestion to perform multivariable regression analysis is very valuable, but it does not fully correspond to the aim of our work, which is to find all, not only independent, factors influencing morphology and retinal vascularity of macula. In addition, carrying out multivariable regression analysis is significantly hampered by the fact that most of the studied variables have significant deviations from the normal distribution. Therefore, the authors decided not to carry out multivariable regression analysis at the moment but we will consider it planning our further study on the bigger sample size.

3. line 172: I would like to see some analysis for the excluded eyes regarding their gestational age, BCVA, etc,......as it is likely that these cases are those with more pathology and poorer VA and less fixation and hence led to selection bias. this needs to be clarified comparing the patients characteristics between included and excluded eyes to exclude bias. 

Reply: We appreciate the Reviewer’s concern. Indeed, we were not able to analyze all examined eyes. In most of the cases GCC (82%) and crossline (72.8%) scans were obtained. However, excluded scans had poor quality and motion artifacts which made reliable analysis impossible. OCT angiography scans require longer fixation and full cooperation of patients therefore we noted higher proportion of excluded eyes (43.1%). In younger children it was difficult to obtain good quality image. However, that does not mean that the examination is not applicable to these children. The cooperation is very important here and lower age might be an issue in terms of OCT angiography scans. It might be stressed that age of excluded eyes was comparable to analyzed eyes regarding GCC and CCT. In individual cases, despite a slight nystagmus and low visual acuity, it was possible to perform good quality examination that were used for analysis.

We have performed comparison between analyzed eyes and the excluded eyes regarding baseline clinical and ophthalmological characteristics as recommended. We have added these results into the manuscript as a supplementary file (Table S1). Interestingly, the statistical results are not fully consistent but same differences between excluded and included cases exist and the selection bias can’t be neglected. We have addressed this issue in the limitation of the study. Nevertheless, the excluded cases are those with more pathology and poorer VA, higher refractive error and less fixation in case of GCC images. Additionally gestational age, birth weight, presence of ROP and treatment played an important role in GCC analysis between included and excluded eyes. It might be expected that wider analysis with excluded eyes might strengthen our result or help to reveal differences also in other examined parameters of OCT/angio OCT rather than cease proven differences between preterm children and healthy controls. Patients with lower gestational age and retinopathy had higher percentage of excluded OCTA images.